# Insights into the molecular basis of tick-borne encephalitis from multiplatform metabolomics

**YanDan Du**[1][⊗], **ZhiHui Mi**[2][⊗], **YaPing Xie**[3], **DeSheng Lu**[1], **HaiJun Zheng**[1], **Hui Sun**[1], **Meng Zhang**[2]*, **YiQing Niu**[1]*

1 Department of Clinical Laboratory, Inner Mongolia Forestry General Hospital (The second Clinical Medical School of Inner Mongolia, University for the Nationalities), Hulunbuir, Inner Mongolia, China, 2 Inner Mongolia Di An Feng Xin Medical Technology Co., LTD, Huhhot, Inner Mongolia, China, 3 SCIEX China Technology Co., Beijing, China

⊗ These authors contributed equally to this work.
* 18047168855@163.com (MZ); Mizh@faith-m.com (YQN)

**Data Availability Statement:** All relevant data are within the manuscript and its Supporting Information files.

## Abstract

### Background

Tick-borne encephalitis virus (TBEV) is the most prevalent arbovirus, with a tentative estimate of 10,000 to 10,500 infections occurring in Europe and Asia every year. Endemic in Northeast China, tick-borne encephalitis (TBE) is emerging as a major threat to public health, local economies and tourism. The complicated array of host physiological changes has hampered elucidation of the molecular mechanisms underlying the pathogenesis of this disease.

### Methodology/Principle findings

System-level characterization of the serum metabolome and lipidome of adult TBEV patients and a healthy control group was performed using liquid chromatography tandem mass spectrometry. By tracking metabolic and lipid changes during disease progression, crucial physiological changes that coincided with disease stages could be identified. Twenty-eight metabolites were significantly altered in the sera of TBE patients in our metabolomic analysis, and 14 lipids were significantly altered in our lipidomics study. Among these metabolites, alpha-linolenic acid, azelaic acid, D-glutamine, glucose-1-phosphate, L-glutamic acid, and mannose-6-phosphate were altered compared to the control group, and PC(38:7), PC(28:3;1), TAG(52:6), etc. were altered based on lipidomics. Major perturbed metabolic pathways included amino acid metabolism, lipid and oxidative stress metabolism (lipoprotein biosynthesis, arachidonic acid biosynthesis, leukotriene biosynthesis and sphingolipid metabolism), phospholipid metabolism and triglyceride metabolism. These metabolites were significantly perturbed during disease progression, implying their latent utility as prognostic markers.

### Conclusions/Significance

TBEV infection causes distinct temporal changes in the serum metabolome and lipidome, and many metabolites are potentially involved in the acute inflammatory response and

**Funding:** This study was funded by grants from Natural Science Foundation of China (No.2017MS0338) and Natural Science Foundation of Inner Mongolia (grant awarded to YDD). The funders had no role in study design, data collection and analysis, decision to publish, or preparation of the manuscript.

**Competing interests:** The authors have declared that no competing interests exist.

immune regulation. Our global analysis revealed anti- and pro-inflammatory processes in the host and changes to the entire metabolic profile. Relationships between metabolites and pathologies were established. This study provides important insight into the pathology of TBE, including its pathology, and lays the foundation for further research into putative markers of TBE disease.

## Author summary

Tick-borne encephalitis virus (TBEV) with extreme contagiousness is a key danger to public health systems in Europe and Asia. To date, little information is obtained about the molecular mechanism underlying infection, and although commercial vaccines against TBEV exist, there is no specific treatment for the disease. Metabolomics and lipidomics offer multiple-visions of metabolome and lipidome sights and help elucidating metabolic to disease phenotype. Serum metabolism and lipidome analysis were performed based on mass spectrometer (MS) platform on a cohort of TBEV patients. About 400 metabolites performed crucial shifts in TBEV patients compared with healthy subjects. This study revealed that in the stage of infection, the host metabolome is tightly regulated, with anti-inflammatory processes modulating pro-inflammatory processes implying the self-limiting phenotype of TBEV and the inherent regulation in humans. The crucial perturbed metabolic pathways contained amino acid metabolism, fatty acid metabolism and phospholipid metabolism. This study provides a powerful and new approach to decipher the interactions between host and virus. These potential metabolites provide high sensitivity and specificity and have the capacity to function as biomarkers for disease surveillance and estimation of therapeutic interventions.

## Introduction

Tick-borne encephalitis (TBE) is a severe central nervous system infection caused by *tick-borne encephalitis virus (TBEV)*, a single-stranded, positive-sense RNA virus of the genus *Flavivirus* [1]. TBE was first reported in Australia and Russia in 1930 [2].TBE is a natural infectious disease that is prevalent in Europe and the northeast of China [3,4] in spring and summer (mainly from March to July). TBEV is divided into three distinct serotypes: Far-Eastern, Siberian and European subtypes. The severity of disease varies among the serotypes [5,6], with the Far-Eastern subtype being more serious and having a higher probability of neurological sequelae and a higher fatality rate than the other two subtypes. The dominant TBEVs isolated from China are of the Far-Eastern subtype. In nature, TBEV is transferred by ticks parasitizing wild vertebrate hosts, primarily small mammals such as rodents and hedgehogs. Large animals, such as deer and horses are non-preferred hosts for TBEV. Over the last few years, TBEV infections occurred most commonly in forest zones or far from city centers. With the continued development of the west of China and rejuvenation of the northeast of the country through ecological tourism, tick activity has increased and tick-borne disease now poses a threat to local residents [7]. Over the past 30 years, the incidence of TBE has increased by almost 400% in Europe, and is rising in China and elsewhere, thus making it a growing public health concern worldwide [8,9]. Progression of TBEV infection generally follows two courses: the first course is characterized by sudden-onset fever accompanied by nausea, fatigue, headache, joint pain and other symptoms, with a duration of about 8 days [10,11]. The second

course of TBE occurs in 20–30% of infected patients and is marked by central nervous system symptoms of varying severity [1].

TBEV can generally be controlled by self-regulation, but its clinical symptoms can be weakening, including considerable brain damage and even death. In a small percentage of patients, TBEV evolves into a more severe form during the acute phase (AP), which is characterized by damage to the central nervous system. At present, no specific diagnostic test or antiviral therapy is available for TBEV. Therefore, it important to reveal the molecular mechanism of TBEV. Both viral and host immune elements appear to be referred, but the roles of each are not entirely understood.

Metabolomics and lipidomics are rapidly emerging fields of 'omics' research that design to quantify and characterize dynamic changes in all metabolites at the system level in response to an internal stimulus or external disturbance [12–14]. Metabolites serve as direct signatures of biochemical activity that are chemically transformed during metabolism and are therefore easier to correlate with phenotype [15]. Lipidomics, a branch of metabolomics, focuses on lipid constituents, which are involved in signaling and infection processes. Cellular homeostasis is disturbed under various disease conditions, and the human body may attempt to maintain homeostasis of the cellular environment through up- or down-regulation of endogenous metabolites. Metabolomics provides a record of the metabolomic status of organisms, and thus offers valuable insight into the mechanisms underlying disease processes. Metabolomics has been applied to infectious diseases to elucidate the interactions between host and virus [16,17]. In recent years, more attention has been paid to lipidomics, including research into how lipids regulate and control the alterations required for viral entry, replication and release [18,19]. To understand the complex and dynamic virus-host interactions of TBEV in vitro, detailed images are essential. However, few omics studies to date have described how the human host responds, biochemically and physiologically, to TBEV and other flaviviral infections, or how it handles other exotic invasions.

Using gas chromatography coupled to mass spectrometry, to identify significant difference in the metabolic profile among uninfected I.scapularis nymphal ticks, B. burgdorferi-infected nymphal ticks and B. mayonii-infected nymphal ticks, and the difference in the abundance of metabolism suggest that different Lyme disease spirochetes may have different metabolic capabilities. The underlying variable metabolites could aid development of novel methods to control spirochete transmission [20]. Villar et al. carried out an integrated metabolomic, transcriptomic and proteomic study that provided an extensive "map" of physiological changes in TBEV patients and characterized the metabolic pathways essential to the tick response to infection, as well as the associated cellular response and molecular mechanisms [21]. Research into tick-borne flaviviruses in the omics era has provided insight into tick biology and pathogen transmission at the genomic level, but more research on tick–virus metabolomics is needed [22,23].Therefore we analyzed serum samples using liquid chromatography quadrupole time-of-flight mass spectrometry (LC-QTOF-MS) to identify the metabolites associated with the human response to TBEV.

In the present study, metabolomic and lipidomic analyses were conducted using untargeted and targeted approaches on serum samples with the purpose of identifying metabolic pathways correlated to disease progression and understanding the mechanisms of TBEV infection. Our results showed that TBEV infection caused significant serum metabolome- and lipidome-wide changes in TBEV patients. Potential metabolites were confirmed via metabolomics and lipidomics, and the results revealed that TBEV infection quickly induced an acute immune and inflammatory response from the host. We propose that changes to the host metabolome and lipidome play crucial physiological roles in the anti-inflammatory and pro-inflammatory processes during the response to tick-borne virus infection to maintain global homeostasis.

## Materials and methods

### Ethics statement

The protocol of this study was approved by the ethics committee of Inner Mongolia Forestry General Hospital, and all eligible individuals provided written informed consent. All information about the participants was anonymized.

### Study design

The study criteria were strictly aligned with the diagnostic criteria for forest encephalitis [24]. The patients were divided into acute phase (AP) and recovery phase (RP) according to their course and clinical manifestations. In brief, adult tick bite patients (>21years) showing acute onset symptoms ($\geq$38.0°C for <72 h, accompanied with nausea, fatigue and joint pain), and without other diseases such as heart, brain, liver, kidney and autoimmune diseases, were included in the study as AP patients. Meanwhile, RP patients presented with an influenza-like illness involving fever, nausea, and fatigue at fever days 3–8. "Fever days" refers to the number of days post onset of fever. Young teenagers were excluded in consideration of differences in the response to TBEV according to age. The diagnosis of forest encephalitis was made based on serology based on an IgG:IgM antibody ratio $\geq$ 1:20. The ratio of IgG:IgM is an indicator of virus infection and the algorithm rule has been adopted in the diagnostic criteria for forest encephalitis. Venous blood samples were collected on admission after tick bite and fever onset. It should be noted that some patients had been treated by themselves or at a clinic before going to the hospital for treatment. The final classification of TBEV was based on clinical test results and symptoms of the 50 TBEV patients that were finally enrolled in this study (May 2018-through September 2019); the patients tested positive for dengue IgG and IgM antibodies in the acute sera using a commercially obtained enzyme-linked immunosorbent assay kit (Pan-Bio, Brisbane, Australia). These patients were deemed to have TBEV infection and were thus included in this study. A detailed hematological analysis was also carried out. Additionally, 39 asymptomatic age-matched healthy subjects participating in an annual hospital staff examination were used as controls. The detailed sample information is listed in **S1 Table**.

### Sample collection and preparation

Serum samples were collected according to the ethical standards of Inner Mongolia Forestry General Hospital. Five milliliters of venous blood werecollected into a blood tube in the morning under fasting conditions and stored without anticoagulant. The whole-blood sample was left at room temperature to clot (~30 min) and then centrifuged to remove the clot. The processed samples were aliquoted and frozen at −80°C.

For metabolomic analysis, 300 µL of each serum sample was thawed at 4°C. Serum samples were deproteinated with 1,200 µL ice-cold acetonitrile in purified water (4:1, V:V). After vortexing, the mixture was centrifuged at 11,000 rpm for 10 min at 4°C. Then, the supernatant was vacuum-dried and re-dissolved with the initial gradient of the mobile phase. To prevent batch effects, assays were conducted in random order. To evaluate the stability of the experimental procedure, quality control (QC) samples were prepared containing equal volumes of all analyzed samples. Eighteen control serum samples were injected randomly in sequence.

For lipidomic analysis, lipids were extracted from serum using a revised version of the Bligh and Dyer method [25]. Briefly, 100-µL serum samples were extracted with 480 µL methanol-chloroform solution (5:1, V:V). After vortexing for 60 min and incubation at 4°C, the sample mixture was centrifuged at 12,000 rpm and 4°C for 15 min. The supernatant was dried and then recovered with 100 µL dichloromethane-methanol (1:1, V:V), vortexed for 30

s and ultrasonicated for 10 min on ice, and then centrifuged at 12,000 rpm and 4˚C for 15 min prior to analysis. The preparation of QC samples and the injection batch processing were conducted as described in the metabolomics procedure above. Six random control serum samples were injected at the start and end of each analytical batch to condition the analytical platform.

## Metabolomic analysis using liquid chromatography-mass spectrometry (LC-MS)

High-performance liquid chromatography time-of-flight mass spectrometry (HPLC-TOF-MS) analysis was performed using a Jasper HPLC system coupled with a TripleTOF 5600 mass spectrometer (AB Sciex, Framingham, MA, USA) equipped with an electrospray ionization source (ESI). Based on polarity, the compounds were separated using an HSS T3 column (1.8 μm, 2.1×100 mm; Waters, Milford, MA, USA) and an Acquity UPLC BEH Amide column (3 μm, 2.1×100 mm; Waters). The oven temperature was set to 40˚C. Analyses were performed in both positive and negative ionization modes over a mass range of 50 to 1200 m/z (mass-to-charge ratio). Both the detailed parameters of the gradient elution program and the MS parameters are listed in **S2 Table**. The stability of the analytical platform and LC-MS method was evaluated using the QC samples. The relative standard deviations of the peak areas in the QC group were below 10%, showing good repeatability and consistency of chromatographic separation throughout the batch.

## Lipidomic analysis using liquid chromatography tandem mass spectrometry (LC-MS/MS)

To identify serum lipid changes, an LC/MS platform was used. A Kinetex C18 column (2.6 μm, 2.1 ×100 mm; Phenomenex, Torrance, CA, USA) operating at 300 μL/min was used to separate 1-μL amounts of lipids into aliquots prior to MS. The column oven temperature was set to 40˚C. The optimized mobile phase consisted of 10 mM NH4Ac in $H_2O$-MeOH-acetonitrile (1:1:1) (solvent A) and10 mM NH4Ac in isopropyl alcohol (solvent B). The gradient was as follows: initial condition of 20% solvent B; 0–3 min, 20–40% B; 3–6 min, 40–60% B; 6–13 min, 60–80% B; 13–17 min, 80–100% B; 17–19 min, 100–100% B; 19–19.1 min, 100–20% B; maintenance of 20% B until the next injection.

## Data pretreatment and statistical analysis

The metabolomics datasets obtained by LC-MS analysis were converted to mzData format via the open-source software MS-Dial 3.82 for peak finding. The output files of the QTOF runs were aligned using various forms of chromatography mass spectrometry (XCMS version 1.26.0), an R-based platform for raw LC-MS data processing and visualization, following retention time correction, data filtering and feature extraction. For MS peak list alignment, the parameters of mass tolerance and retention time (RT) tolerance values were set to 0.25 Da and 30 s, respectively. The aim of this procedure was to reduce the shift in RT and avoid redundant signals. The algorithm and parameters used for peak detection were as follows: minimum peak width, 5 s; maximum peak width, 20 s; ppm deviation, 5 ppm; signal-to- noise threshold (snthresh), 4. Then, the report table was imported into MetaboAnalyst (version, 4.0), a set of online software packages for multivariate data analysis. Sum, log-transformation, and pareto-scaled algorithms were sequentially applied for normalization to reduce any systematic bias and improve overall data consistency to ensure that biological comparisons could be obtained. Principal component analysis (PCA) was used as an unbiased statistical method to observe the

clustering situation of healthy controls and TBEV patients. PCA analysis was used to confirm the quality of the datasets and to visualize differences between groups of samples in the unsupervised analysis. To achieve the distinction between TBEV patients and the control group, supervised multivariate analysis and orthogonal partial least squares data analysis (OPLS-DA) were employed to identify potential metabolites. The selecting algorithms for potential differential metabolites were based on variable importance in projection (VIP) values; variables with a VIP>1 made a significant contribution to the separation of samples in the OPLS-DA analysis. Student's t-test was performed to verify whether the differential metabolites acquired from OPLS-DA model were statistically significant among groups ($p <$ 0.05). In addition, the fold change (FC) threshold was also used as an indicator to differentiate the groups.

For lipidomics datasets, information-dependent acquisition (IDA) coupled with dynamic background subtraction was used to obtain comprehensive MS and MS/MS spectra and to automatically deduct the background spectrum to ensure the validity of the data. The commercial software LipidView and SCIEX OS (AB Sciex) were employed to extract signals and further discriminate and quantify lipids. The statistical analysis followed the protocol for metabolomics data. The detailed workflow is shown in **S3 Table**.

## Metabolite identification

The identification of lipid molecules and metabolites followed an established strategy[26]. The SWATHtoMRM method first captures the profile from one pooled biological sample and then acquires the MS/MS spectra for all metabolites, increasing the credibility of biomarker identification. The identification process is described here using phosphatidylethanolamine (PE) as an example. First, the elemental composition $C_{39}H_{75}NO_8P$ was calculated for the m/z 716.52 ion based on its exact mass, the nitrogen rule and the isotope pattern. Then, the elemental composition and exact mass were used to search open-source databases, including the METLIN Metabolite and Chemical Entity Database (https://metlin.scripps.edu/), the Kyoto Encyclopedia of Genes and Genomes (KEGG; https://www.kegg.jp/), and an in-house database. Next, MS/MS spectral information was obtained to assess fragmentation of the metabolite. Taking PE for example, m/z 141.02 is the characteristic neutral loss ion of PE, and m/z 255.2327 and m/z 279.2319 are the daughter ions of PE; their presence enhanced confidence in the identification. An example of the identification procedure of PE (34:2) as a differential is shown in **S1 Fig**. Although the identification we obtained was putative since most markers were searched without authentic standards, the program SWATHtoMRM acquired MS/MS spectra for all samples, then extracted a set of multiple-reaction monitoring transitions for targeted analysis, demonstrating the advantages of this technology.

## Pathway analysis and interaction networks

To comprehensively examine the metabolomics profiles, all relevant detected metabolites were mapped to metabolic pathways using MetaboAnalyst 4.0 (www.metaboanalyst.ca) [27]and the LIPID MAPS Lipidomics Gateway (http://www.lipidmaps.org/) [28]. The enrichment analysis was performed using a topology analysis that takes into account the positions of significant metabolites in metabolic pathways. Two indexes were estimated in the topology analysis: the centrality and the pathway impact. The centrality measures the number of shortest paths going through the target metabolite, which is employed to assess the significance of each metabolite within a specific pathway. The pathway impact is the cumulative value of the significant metabolites.

## Results

### Serum metabolomics and reproducibility of the analysis

The serum metabolic markers obtained through metabolomics and lipidomics are displayed in S2 Fig. The filtered and normalized datasets obtained via LC-MS analysis were used to create PCA models to visualize the sample groupings (Fig 1). The reliability of the experimental procedures and the stability and reproducibility of the analysis were determined using QC samples, as shown in the PCA plots. The cluster of QC samples verified the reliability of the analysis method by showing that it did not affect the grouping of samples; different sample groups should be biologically distinct. PCA scores representing TBEV samples and healthy samples were obtained using the T3 column and BEH Amide column; the PCA results obtained through lipidomics are shown in S3 Fig.

### Global metabolomic changes with TBEV infection

For a comprehensive understanding of metabolite coverage during TBEV infection, both the T3 and BEH amide columns were used to obtain a wide range of metabolites. The T3 column focuses on metabolites with low polarity, and the amide column on compounds with high polarity. PCA and OPLS-DA mode were used to compare the experimental and control groups. The PCA score plot (Fig 1) was used to visualize metabolomic changes in TBEV patients. Healthy controls (n = 39), patients in RP (n = 31) and patients in AP (n = 19) were compared. The strongest response of patients to TBEV infection occurred in the first 4 days of infection, and homeostasis recovery (RP) appeared around the fourth day of fever, accompanied by mild clinical symptoms, such as fever and headache; the AP accompanies with the course of encephalitis, which has been observed to come to a head during early febrile phase. PCA showed that healthy and disease groups were well distinguished, but the unique metabolome profile between AP and RP was not revealed, suggesting the complexed duration time of illness, and a majority of patients recover, which could hint the metabolome similarities between patients at AP and RP in the PCA score plot. The small number of samples also contributed to the phenomenon. Partial least squares data analysis (PLS-DA) and OPLS-DA, a rigorous supervised analysis method, revealed distinct separation of the healthy controls and

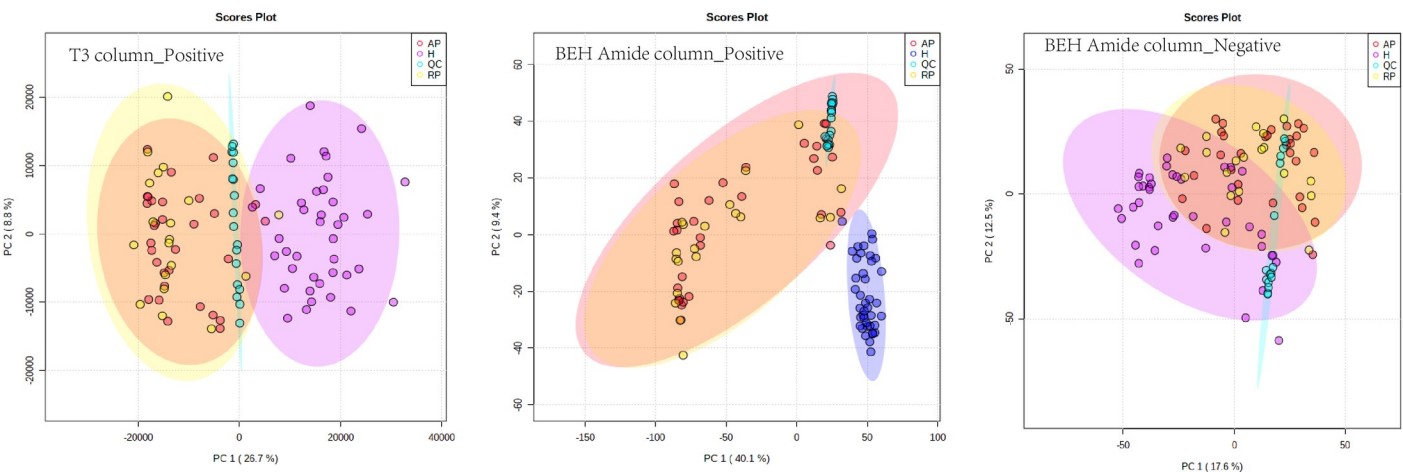

**Fig 1. PCA models created using data obtained through LC-ESI-TOF-MS analysis in positive ionization mode on a T3 column (PC$_1$ = 26.7%, PC$_2$ = 8.8%), amide column (PC$_1$ = 40.1%, PC$_2$ = 9.4%), and in negative mode on an amide column (PC$_1$ = 17.6, PC$_2$ = 12.5%).** Red, purple, blue, and yellow solid circles correspond to AP, healthy, QC, RP patients, respectively.

TBEV patients; however, AP patients and RP patients were overlapped. The OPLS-DA algorithm was adopted to analyze the degree of separation of patients in two courses (S5 Fig). The usage of $R^2Y$ and $Q^2$ parameters evaluated the quality of the OPLS-DA model to show the fitness and predictive ability of the model, respectively. The $R^2$ and $Q^2$ values of the OPLS-DA data were both $<0.3$, suggesting the imperfect performance of the model. Generally, $R^2(Y)$ and $Q^2$ values of $>0.4$ are acceptable limits for clinical samples. Therefore, in subsequent experiments, we focused mainly on the control group and an overall disease group. In future studies, we aim to collect more samples to enable grouping based on disease course, to aid in elucidating the mechanism of TBEV development.

## Identification of significantly altered metabolites and pathways

Approximately 529 differential metabolites were identified via LC-MS using T3 and amide columns, in positive and negative mode; 181 and 182 features of those metabolites were identified on the amide column in positive and negative mode, respectively, and 166 were identified on the T3 column in positive mode (S5 Table). After eliminating repeats, 440 metabolites were retained. Based on the criteria- of a VIP value$>1$ in the OPLS-DA and a *p value* $<0.05$ on the *t*-test, 144 of these metabolites were structurally identified with LC-MS/MS analysis, based on discriminant analysis of the healthy control and TBEV groups. The metabolites belonged to classes such as acylcarnitine, free fatty acid (FFA), free amino acid (FAA) and their derivatives, phosphatidylcholine (PC), purine, sphinganine and carbohydrate. Detailed information is listed in S6 Table. These potential biomarkers showed different trends; some exhibited increased expression, while others showed decreased expression. The metabolites in the TBEV group showed greater changes compared to those of the control group, indicating that they might serve as potential biomarkers of viral infection and thus could help to elucidate the pathological mechanism.

Interestingly, metabolites within the same category generally showed similar trends. Amino acids, including D-glutamine, pyroglutamic acid, and L-cystine, showed a downward trend, consistent with a previous report of a total reduce in plasma amino acid levels (S6 Fig)[11,29]. Leukotriene (LT) A4 and arachidonate levels were elevated in TBEV samples. These metabolites are involved in lipoxin biosynthesis which is in turn involved in the host immune response. Sphingomyelin (SM) metabolism intermediates, such as sphingosine 1-phosphate, sphinganine 1-phosphate, sphingosine and serine showed decreased levels in TBEV patients (S7 Fig).

We used MetaboAnalyst to integrate our datasets and reveal the pathways in which the identified metabolites are involved. The data revealed that the dominant disordered metabolic pathways included amino acid metabolism, sphingolipid metabolism, lipoxin biosynthesis and aspirin-triggered resolvin E biosynthesis. The metabolite network manifested that the most vital changes occurred in lipid metabolism and energy supply pathways; the major altered pathway are summarized in Fig 2.

## Global serum lipid changes with TBEV

PLS-DA revealed obvious serum lipidome changes in the TBEV group (S4 Fig). To identify potential lipid biomarkers and intermediates for differentiating the healthy control and disease groups, we referred to the OPLS-DA score plot. PCs, SMs and triacylglycerols (TAGs) were best able to distinguish the groups (Table 1). We observed clear negative correlation between PC and TAG expression levels. The downward trend of SM levels in TBEV patients was dominated by TAG60:10|TAG 18:0_20:4_22:6. All of the SMs identified were decreased in the disease group (Fig 3).

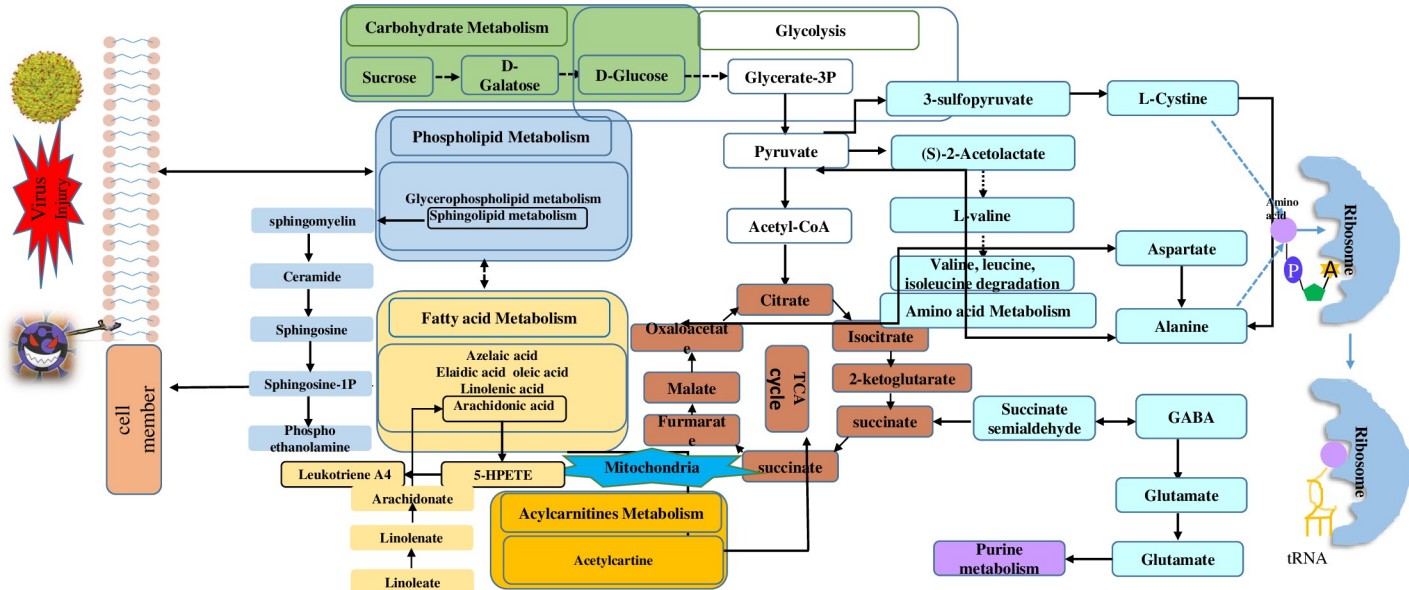

**Fig 2. Major altered lipid and energy metabolic pathways in TBEV patients.**

**Table 1. Potential biomarkers between TBEV patients and the healthy control group.**

| Lipid | *p*-value | Log2 (FC) | VIP | Trend |
|---|---|---|---|---|
| PC 36:5\|PC 16:0_20:5 | 1.44E-14 | 10.54 | 6.43 | ↑ |
| PC 28:3;1 | 4.33E-17 | 1.89 | 2.18 | ↑ |
| PC 36:2;1 | 1.44E-12 | 8.07 | 5.03 | ↑ |
| PC 38:3;2 | 1.60E-11 | 1.21 | 1.53 | ↑ |
| PC 38:7 | 9.07E-15 | 9.67 | 6.48 | ↑ |
| CE 18:3;1 | 4.4266E-19 | 3.65 | 3.15 | ↑ |
| LPE O-16:1 | 2.88E-09 | 1.24 | 1.57 | ↑ |
| TAG 60:10\|TAG 18:0_20:4_22:6 | 0.001 | −1.08 | 1.01 | ↓ |
| PE O-37:5\|PE O-17:1_20:4 | | | | |
| TAG 60:11 TAG 16:0_22:5_22:6 | 2.46E-05 | −1.10 | 1.03 | ↓ |
| DAG 40:8\|DAG 18:2_22:6 | 1.09E-05 | −1.06 | 1.25 | ↓ |
| PC 30:1 | 5.44E-05 | 1.02 | 1.27 | ↑ |
| PC 38:3;2 | 3.02E-06 | 1.04 | 1.18 | ↑ |
| PC 28:3;1 | 4.23E-13 | 2.04 | 2.15 | ↑ |
| PC 36:5\|PC 16:0_20:5 | 8.23E-39 | 10.54 | 6.51 | ↑ |
| PC 38:7 | 4.84E-41 | 9.83 | 6.41 | ↑ |
| CE 18:3;1 | 6.94E-18 | 3.68 | 2.93 | ↑ |
| TAG 42:0\|TAG 10:0_16:0_16:0 | 0.012673 | 1.20 | 1.17 | ↑ |
| TAG 52:6 | 0.001648 | −1.03 | 1.12 | ↓ |
| LPE O-16:1 | 5.01E-9 | 1.34 | 1.63 | ↑ |
| PC 36:2;1 | 3.02E-06 | 7.92 | 4.77 | ↑ |
| TAG 54:7(18:2/18:2/18:3) | 0.001 | −1.17 | 1.13 | ↓ |
| TAG 54:8\|TAG 18:2_18:3_18:3 | 0.005 | −1.66 | 1.44 | ↓ |
| TAG 58:10(18:2/18:2/22:6) | 0.0001 | −1.04 | 1.14 | ↓ |
| TAG 59:7 | 0.0001 | −1.22 | 1.37 | ↓ |
| DAG 40:8\|DAG 18:2_22:6 | 6.56E-05 | −1.08 | 1.20 | ↓ |

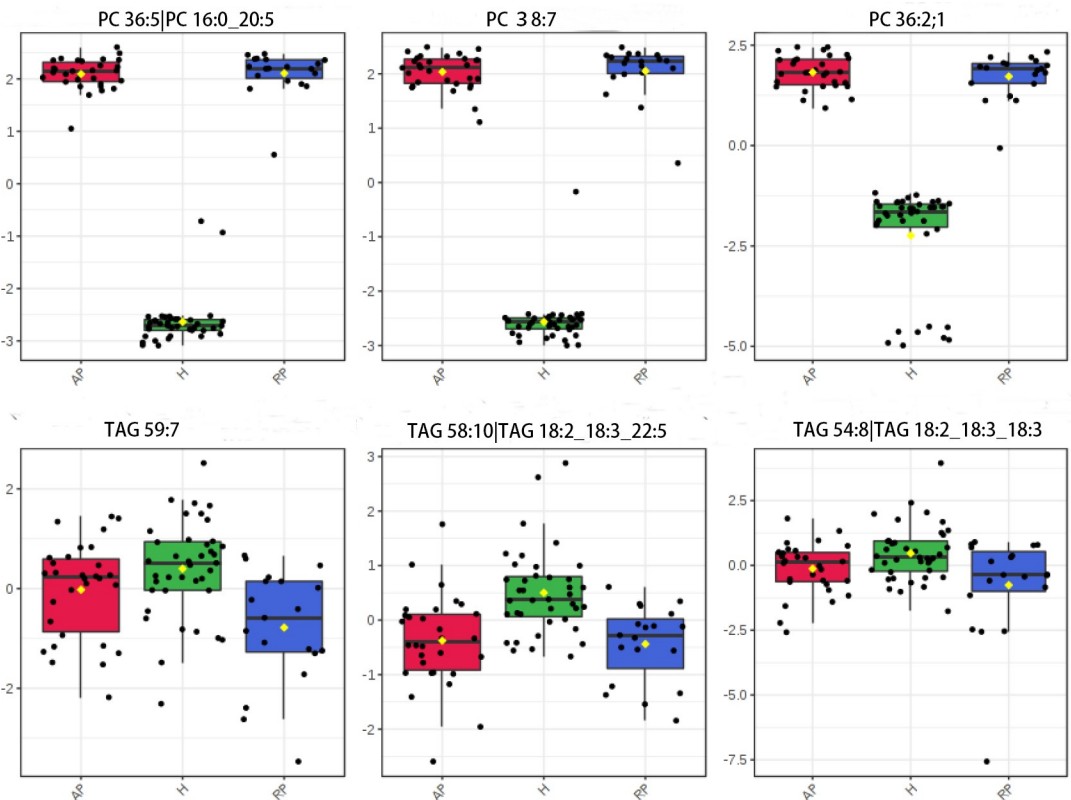

**Fig 3. Relative abundances of differential metabolites in serum samples collected from patients and controls based on lipidomics.** (red box: acute phase patients; blue box: recovery phase patients; green box: control group).

Dominant pathway analysis revealed nine metabolic pathways potentially related to TBEV (**Fig 4**). Four of these pathways are closely linked to lipid metabolism (arachidonate metabolism, LT metabolism, lipoprotein biosynthesis and sphingolipid metabolism), and two others to phospholipid metabolism and triglyceride metabolism. The other three (D-glutamine metabolism, pyroglutamic acid metabolism and L-cystine metabolism) are closely associated with amino acid metabolism. A graphical representation of the pathway analysis results is shown in **Fig 4**. A comprehensive schematic diagram of the TBEV-induced network of molecular interactions was constructed, which shows the relationships among the identified metabolites (**Fig 2**).

## Discussion

TBEV is a self-limiting febrile illness from which complete recovery is generally made, with little to no sequelae, which clearly suggests effective protective mechanisms in humans against TBEV. In a few serious cases, TBEV causes nerve damage in the brain, inducing encephalitis. Because of the influence of heterotypic infection, the unraveling of protective mechanism is complex and difficult to understand. The study of recovery from TBEV a physiological point of view is therefore crucial in deepening our understanding of how humans defend against and recover from TBEV infection and potentially other flavivirus-induced illnesses. In this study, metabolomics and lipidomics were applied to understand the global metabolomic changes that occur in TBEV patients, through which we identified metabolic pathways that are altered during TBEV infection. The sample cohort size (59 patients and 30 healthy controls) at

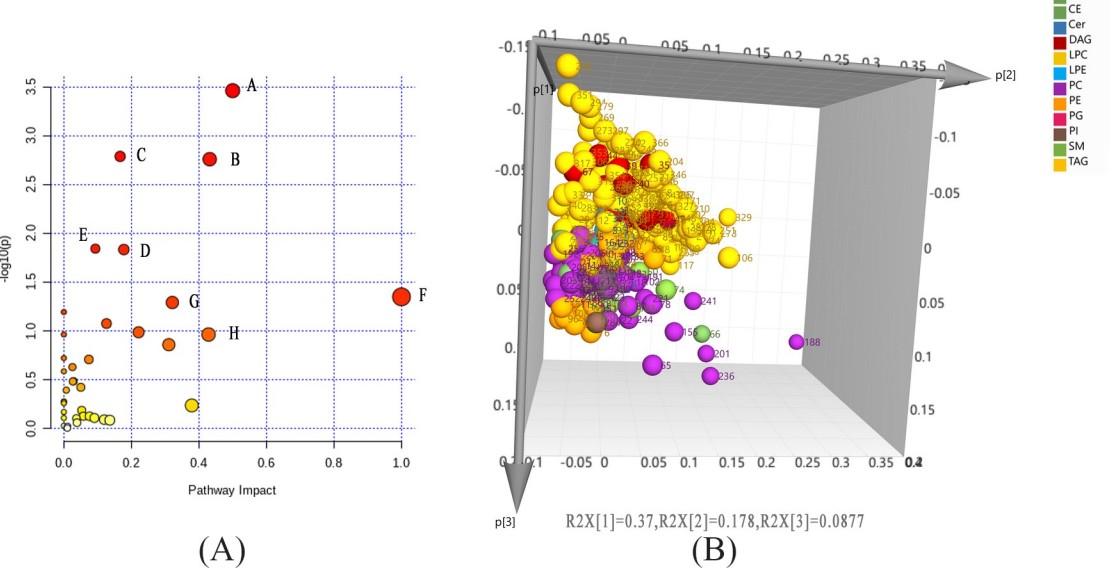

**Fig 4. Graphical representation of the pathway analysis of significantly altered metabolites in the three groups (A: metabolomic results; B: lipidomic results).** A:D-glutamine and D-glutamate metabolism; B: glycine, serine and threonine metabolism; C: aminoacyl-tRNA biosynthesis; D: arginine metabolism; E: sphingolipid metabolism; F: linoleic acid metabolism; G: arginine and proline metabolism; H: taurine and hypotaurine metabolism. LPC: lysophosphatidylcholine; SM: sphingomyelin; TAG, triacylglycerol. Dark bule circle: cluster of LPC metabolites; red circle: cluster of SM metabolites; light blue circle: cluster of TAG metabolites.

two-dimensional time points of infection, including AP and RP patients, seized dynamic changes in metabolites and the excavation of reliable differential information that were closely related with TBEV pathophysiology. By using multiple separation methods, we found that approximately 500 serum metabolites were significantly changed, of which 144 were identified by LC-MS/MS. Lipids, which constitute a biochemically important subclass of metabolites, were also studied and verified. These metabolites included a number of pro- and anti-inflammatory mediators, indicating a rapid response by the host to excessive inflammation to prevent further damage from TBEV infection. The altered metabolites may serve as diagnostic markers of the disease and provide insights into its progression. Although these proof-of-concept diagnostic metabolites are encouraging, they were identified based on a small number of samples; additional studies with more patients are needed to confirm the results. It must also be noted that the results were restricted to non-pediatric patients from the local area. Further studies will be necessary to determine if these (and other) metabolites are predictive of the progression from RP to AP in different patient populations.

PCA revealed surprisingly weak clustering of samples from RP and AP TBEV patients (**Figs 1 and S3**). Many factors have been shown to affect TBEV disease severity, including TBEV serotype, geographical latitude and individual sample differences[7,30,31]; such factors could have confounded our analysis, contributing to the lack of clear clustering.

These results provide proof-of-concept that differential perturbation of the serum metabolome is associated with TBEV infection and disease outcome, and that the trends of altered metabolites are associated with TBEV type (AP or RP). However, in this retrospective proof-of-concept study, no detailed serological information was collected to facilitate grouping of the TBEV patients according to disease status (**Figs 1 and S3**). Thus, the metabolites identified in this retrospective study may represent metabolic perturbations after TBEV infection, rather

than metabolic phenomena reflecting disease status. Despite the small sample size, 144 metabolites differentiated the TBEV patients from the healthy group, including structurally confirmed metabolites involved in amino acid metabolism, lipoxin biosynthesis and aspirin-triggered resolvin E biosynthesis. Twenty-eight biomarkers with an area under the receiver operating characteristic curve (AUROC) value >0.9 were identified (**S4 Table**); these candidate biomarkers are not specific to TBEV disease, but when combined with a TBEV-positive serological test (eg., IgG:IgM antibody ratio>1:20) may be useful for diagnosing and predicting the prognosis of TBEV infection based on serum specimens.

A few recent studies have applied metabonomics approaches to discover new diagnostic and prognostic markers for TBEV and to understand the mechanism of disease development. Transcriptomic and proteomic analyses have been used to identify metabolic pathways, and some investigations have focused on serological testing[20,21,32,33].

TBEV infection in humans causes early and significant immunological reactions that are initially pro-inflammatory, and then anti-inflammatory[34]. While the acute inflammatory reaction is a prerequisite for initiating the pathogen-killing process, the accompanying anti- and pro-inflammatory processes are vital, as they prevent excessive pathological injury to the host. Many biochemical reactions take place throughout the body, along with the response to anti- and pro-inflammatory processes.

Arachidonic acid (AA) is liberated from cell membrane phospholipids through phospholipid hydrolysis of AA -derived compounds such as prostaglandins, LTs and thromboxanes, which are crucial mediators of the inflammatory response and participated in regulating both the response intensity and durability of the inflammatory response [35,36]. In contrast to the influence of AA, omega-3 polyunsaturated fatty acids (PUFAs), such as docosahexaenoic acid (DHA) and its derivate (4Z,7Z,10Z,13Z,16Z,19Z)-docosahexaenote, are known for their anti-inflammatory response, which inhibits the formation of inflammatory precursors [37,38].Our results indicated an ascending trend of pro- and anti-inflammatory PUFAs during the pyretogenic phase of TBEV infection, including both AA and DHA. Such positive effects of active resolution, the early AA-related inflammation is subsequently resolved by DHA-derived anti-inflammatory media and inhibiting the production of pro-inflammatory cytokines, is deemed to maintain human health and tissue homeostasis, in which AA also regulates and controls the flexibility and fluidity of cell membranes, serves as a lipid second messenger in cellular signaling, acts as an inflammatory intermediate and induces vasodilatation [39,40]. As documented previously, oxidative stress is also associated with AA metabolism [41]. The biosynthesis of LTs, which are derived from AA, occurs through two steps involving 5-lipoxygenase. LTs are inflammatory mediators that play a role in normal host defense; they have also been found to be markers of disease outcome[41,42]. Upregulated or downregulated LT expression and an excessive response to LTs contribute to disease pathophysiology. The downregulation of LTs in TBEV patients revealed decreased defense ability in host cells. We observed lower levels of SM in TBEV patients versus the control group. SM plays important roles in cell membranes and is crucial to human metabolism. Barceló-Coblijn et al. demonstrated that in some cancer cells, the endogenous level of SM is reduced compared to non-malignant cells [43]. Cortisol, which also has powerful anti-inflammatory and immunosuppressive properties, a cytokine-activated glucocorticoid possessing the ability of anti-inflammatory and immunodepressive properties, also showed an elevated trend in the febrile stage of TBEV. Viral infections have the ability to induce an increase in serum cortisol levels during TBEV infection; cortisol levels at the pyretogenic stage were notably higher than in the control group. To inhibit the production of cytokines and other pro-inflammatory mediators is the effective way to control inflammatory and immune responses which may impair the immune response and protect patients with TBEV from cytokine-mediated damage[44].The increase of serum inosine in the

pyretogenic stage was consistent with the increase in oxidative stress of TBEV in the inflammatory state and the change in extracellular inosine concentration[45]. Inosine has been verified to generate a wide range of anti-inflammatory cytokine production including the enhanced production of interleukins (ILs)4, -6, and 10 [46]. In general, our metabolomics data sets show a dynamic metabolite flow in TBEV where acute inflammation is controlled, weakening the immune response and maintaining homeostasis.

On the one hand, the mechanism of immune system for host immunity has been the subject of TBEV pathogenesis. Dendritic cells and B-lymphocytes belonged to immune cells serve dominating roles in viral clearance and as objects of TBEV infection. We explored the inherent relationships between SM concentrations and immune cells and found multidimensional early molecular characters of sphingolipids in mediating the host response to TBEV. The core of sphingolipid enzymes is acidic sphingomyelinase, which has been involved in multiple immune responses including entry of measles virus dendritic cells [47] and exocytosis of cytolytic granules by T cells [48]. Our finding that sphingolipids are linked with lymphocytes during TBEV which may further clarify the importance of B-cell responses and merits deep research. Considering their roles in platelet activation, the lacking of phosphatidic acid and its lyso derivatives was intriguing. Evidence for altered phospholipid metabolism in TBEV patients, in particular the expression of SM, was found in our study. In a study of rhesus macaques with simian immunodeficiency virus-induced central nervous system disease, a global approach provided deep insight into cerebrospinal fluid [49].

On the other hand, simultaneously, pathogenic invasions to the host may be the other element of the underlying mechanism. Extensive changes in serum eicosanoid (e.g., leukotriene A4) and linolenic acid levels were detected. A most common FFA, linolenic acid also showed significant changes to virus infection (e.g., H1N1 and dengue) [50]. Another plausible distinctive metabolome change is the xanthinine derivations. Xanthinine functions to elicit anti-inflammatory immunomodulators that attenuate the damaging effects of the host response [50]. Inosine and hypoxanthine levels are elevated in the febrile period, during which systemic inflammation may need to be taken into account[51,52].The same trend was also confirmed in the chronic systemic inflammation model; it is noteworthy that similar phenomenon are noticed in infections disease where acute inflammation occurs. In the process of inflammation, the increase of adenosine to inosine editing in cytokine mRNA may affect on the degradation of transcripts, supporting the necessary early pro-inflammatory and decomposition reactions in the stage of TBEV. This may be embodied in a fusion of pro- and anti-inflammatory cytokines such as IL-17A, interferon gamma, tumor necrosis factor alpha, IL-4 and IL-8 that show increased levels during the febrile phase.

In addition to the aforementioned metabolite changes in the host during virus invasion, changes in certain metabolites may be more common in different pathogen infections.

Łuczaj et al. demonstrated perturbation of the lipid profile of TBEV patients during the course of infection [53]. In this study, multiplatform metabolomics and advanced statistical methods were utilized to evaluate the serum metabolomic signature of TBEV. The metabolites showing significant alterations were mainly involved in pathways related to amino acid metabolism, lipoprotein biosynthesis, AA biosynthesis, LT biosynthesis, and sphingolipid and TAG metabolism. Some altered metabolites are linked to lipid metabolism and the regulation of inflammatory processes via the control of fatty acids and phospholipids, while others are associated with immune regulation, cell apoptosis and the maintenance of cellular homeostasis [54–56].

TBEV replication is dependent on host cell lipid biosynthesis and metabolism. Viral replication complexes are present in subcutaneous tissues[57]. Dendritic cells in the skin are thought to be the initial cells involved in replication, by transporting the virus to nearby

draining lymph nodes via the lymphatic system. After replication in the lymphatic organs, TBE virus spreads through efferent lymphatics and the thoracic duct resulting in viremia [58]. Phospholipids are normally minor constituents of cell membranes and include C16, C28, C36 and C38 unsaturated acyl chains, the levels of which were elevated in our TBEV group relative to the control group. Phospholipids are precursors of lipid mediators, such as platelet activating factors (PAFs) and eicosanoids, which are involved in inflammatory responses [16,59,60]. Increased phospholipid biosynthesis is reasonable given that host cell phospholipid metabolism is influenced by TBEV replication in both tick and human cells. These single fatty acid chain lipids are involved in alterations of membrane structures, mediation of acute inflammation and regulation of pathophysiological events, throughout the vasculature and at local tissue sites[21,61,62]. Interestingly, PCs play key roles in membrane-mediated cell signaling. Due to their presence in the cell membrane, PCs may disrupt homeostasis of the vascular endothelium, causing physiological and pathological changes, as well as impairing barrier function [63]. Recent studies have shown that lipid metabolism is related to various functions in living organisms. For example, disordered lipid metabolism has been shown to be involved in the pathogenesis of several human diseases, including obesity, cancer and viral diseases [64,65].

Our results showed TAG (TAG16, TAG18, TAG54, TAG59 and TAG60) expression during the febrile stage of TBEV infection. As the main energy reserves of the human body, TAGs are involved in metabolic processes that determine the rate of fatty acid oxidation, the biosynthesis of lipid molecules, and the metabolic fate of lipoproteins. Viral infection activated intracellular biochemical reactions with high energy demands, resulting in the lipolysis of TAG. The pool of endogenous TAGs provides a constant source of fuel for mitochondrial β-oxidation and biochemical reactions in the body.

In the present study, we found that phospholipid levels were significantly increased in TBEV plasma, while TAGs showed the opposite trend. TAGs serve as energy reserves to fuel dynamic metabolic processes. The metabolism of TAGs involves both intracellular and extracellular mechanisms[66]. The metabolism of TAGs is activated by TBEV invasion of the human body, so TAG levels were monitored in our study.

TBE also affects energy metabolism. In our study, the level of acylcarnitine was increased in the disease group. Acylcarnitine plays crucial physiological roles in lipid metabolism, and particularly in fatty acid β-oxidation (FAO), which is needed to transport long-chain fatty acids into mitochondria for lipid oxidation. Changes in serum acylcarnitine levels may indicate mitochondrial dysfunction, which is associated with a disturbance in fatty acid transport [67]. Elevated levels of acylcarnitines suggest an impairment of FAO in TBEV patients, similar to the impaired tricarboxylic cycle seen in an endothelial cell line infected with dengue virus DENV[68]. Notably, such alterations may not be unique to pathological conditions induced by a virus. Using a rat model, Wu et al. demonstrated that these changes may also be related to the development of hypertension [69]. On the other hand, other acylcarnitines are pro-inflammatory cytokine inducers closely associated with inflammation [70].

Some amino acids are useful as biomarkers of the response to pathogenic invasion. In this study, increased activity in the D-glutamine and D-glutamate metabolism pathways, as well as increased alanine, aspartate and glutamate metabolism, and valine, leucine and isoleucine biosynthesis, were observed. Porcheray et al. demonstrated that the concentration of glutamate decreased during viral replication, and also investigated the consequences of human immunodeficiency virus (HIV) infection on glutamine synthetase; this key enzyme showed functional expression in macrophages[71]. Glutamate metabolism is sensitive to both HIV infection and inflammation, and thus could be a therapeutic target in HIV encephalitis[72]. Taurine, a sulfur-containing amino acid and organic osmolyte involved in various physiological processes, exhibits anti-oxidative and anti-inflammatory activities[73]. A relationship between

phenotypes and molecular mechanisms was identified in a mouse model of taurine transporter deficiency, indicating the role of taurine in maintaining physiological homeostasis[73]. Therefore, we hypothesized that serum taurine depletion during TBEV infection may lead to inflammation and affect cellular homeostasis. Serum concentrations of leucine and isoleucine were significantly lower in our TBEV patients relative to the control group. This suggests enhanced cellular demand for these metabolites, as viruses infecting humans require the host body to produce substances that support viral replication and proliferation. A study focused on changes in metabolites in patients undergoing tumor resection showed that leucine and isoleucine are involved in disordered cellular growth [74].Cho et al. elucidated the mechanisms underlying metabolic responses to viral hemorrhagic septicemia virus (VHSV) infection in olive flounder, and suggested that the metabolism of amino acids such as leucine and isoleucine was suppressed [75]. Cao et al. analyzed alanine, aspartate and glutamate metabolism in primary hepatocellular carcinoma (HCC) tumors from alcoholic liver disease (ALD), hepatitis B virus (HBV)-infected, and hepatitis C virus (HCV)-infected cirrhotic patients, and found that the metabolism of all three of these amino acids was useful for differentiating the ALD patients from the HCV and HBV patients, demonstrating that the metabolic phenotypes of primary HCC tumors vary significantly across ALD, HBV-infected, and HCV-infected cirrhotic patients [76].

Aminoacyl-tRNA biosynthesis was observed after infection in our study. Aminoacyl-tRNA biosynthesis may participate in biochemical processes that stimulate synthesis of the viral proteins required for replication. Thus, the present study highlighted the importance of aminoacyl-tRNA biosynthesis in TBEV infection.

## Conclusions

The metabolic changes observed in this study may improve our currently limited understanding of the molecular basis of TBEV. Conventional metabolomics coupled with lipid metabolomics and advanced statistical analysis appears to be a robust approach for multiparametric analysis of TBEV infection, to identify both diagnostic and prognostic markers. This is the first study to use lipid metabolism data to illustrate the interaction between TBEV and its host, and provides new insights into TBEV. However, integrating metabolomics results with those obtained using other omic platforms, such as proteomics, and transcriptomics, is crucial for comprehensive evaluation of the pathological mechanisms of TBEV. In future studies, more samples and integration of genetic and metabolomics data would be useful to detect more alternations that may be unique to TBEV. This study had some notably limitations. First, the sample sizes, especially for the AP group, were small. In addition, the serological data were incomplete. To obtain reliable diagnostic and prognostic biomarkers, further studies are needed. The metabolic biomarker candidates identified here must be validated using larger sample sizes and more advanced analytical techniques.

## Supporting information

**S1 Table. Sample grouping schedule of healthy subjects and TBEV patients enrolled in this study.**
(DOCX)

**S2 Table. Detailed LC-MS information.**
(DOCX)

**S3 Table. Experimental workflow for biomarker discovery.**
(DOCX)

**S4 Table. Biomarkers identified by LC-MS-MS with an AUROC > 0.9.**
(DOCX)

**S5 Table. Identified metabolites of TBEV serum from LC-MS analysis based on T3 and BEH amide columns in positive and negative mode.**
(DOCX)

**S6 Table. Metabolites showing significant changes, according to uni- and multivariate statistical analysis, in the serum samples of TBEV patients compared to healthy controls.** MSI Level 2: Metabolites identified by LC-MS and spectrum similarity with public/commercial spectrum libraries.
(DOCX)

**S1 Fig. Workflow for identification of putative metabolites in serum samples.** Example identification of PE 34:2 as a differential metabolite to expound the authentication process. **A.** Extracted ion chromatogram (EIC) and matched ion m/z 716.5224. **B.** MS/MS spectrum and proposed pathways in positive mode. **C.** MS/MS spectrum and proposed pathways in negative mode.
(TIF)

**S2 Fig.** Chromatograms representing serum metabolic fingerprints of quality control (QC) samples obtained with LC-TOF-MS in (A) positive ionization mode on a T3 column, (B)positive mode on an amide column, (C) negative mode on an amide column and lipidomics in (D) positive mode and (E) negative mode.
(TIF)

**S3 Fig.** PCA models built on lipid data obtained with LC-ESI-TOF-MS analysis in (A) positive mode ($PC_1$ = 22.7%, $PC_2$ = 20.2%) and (B) negative mode ($PC_1$ = 43.1%, $PC_2$ = 14.1%). Red, purple, blue and yellow solid circles correspond to AP, healthy, QC and RP samples, respectively.
(TIF)

**S4 Fig.** PLS models based on metabolomics data obtained with LC-ESI-TOF-MS analysis in (A) positive mode on a T3 column, (B) positive mode on amide column and (C) negative mode on amide column. Red, purple, blue and yellow solid circles correspond to AP, healthy, QC and RP samples, respectively.
(TIF)

**S5 Fig. OPLS-DA score plots of TBEV based on metabolomics data.** The OPLS-DA models were constructed using LC-MS/MS data from acute phase and recovery phase TBEV patients on a T3 column and an amide column.
(TIF)

**S6 Fig. The relative amounts of D-glutamine, pyroglutamic acid, and L-cystine in TBEV patients and the control group.** (Red box: healthy samples; green box: QC samples; blue box: acute phase patients; Modena box: recovery phase patients).
(TIF)

**S7 Fig. Relative amounts of leukotriene (LT) A4, arachidonate and sphingosine in TBEV patients and the control group.** (Red box: healthy samples; green box: QC samples; blue box: acute phase patients; Modena box: recovery phase patients
(TIF)

## Acknowledgments

The authors thank the SHANGHAI Mass Spectra team of AB SCIEX and engineer YaPing Xie of the Beijing Application Centre for providing testing services. The authors also thank all volunteers for their enthusiastic support of our research.

## Author Contributions

**Conceptualization:** YanDan Du.

**Data curation:** DeSheng Lu.

**Formal analysis:** ZhiHui Mi.

**Funding acquisition:** YanDan Du.

**Investigation:** YiQing Niu.

**Methodology:** YaPing Xie.

**Project administration:** ZhiHui Mi, Meng Zhang.

**Resources:** YanDan Du.

**Software:** YaPing Xie.

**Supervision:** HaiJun Zheng, Hui Sun.

**Validation:** ZhiHui Mi.

**Visualization:** ZhiHui Mi.

**Writing – original draft:** YanDan Du, ZhiHui Mi.

**Writing – review & editing:** Meng Zhang, YiQing Niu.

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
