## [Decision Letter · Decision Letter 0]

24 Aug 2020

Dear Dr YiQing Niu,

Thank you very much for submitting your manuscript "Insights into the molecular basis of tick-borne encephalitis from Multiplatform Metabolomics" for consideration at PLOS Neglected Tropical Diseases. As with all papers reviewed by the journal, your manuscript was reviewed by members of the editorial board and by several independent reviewers. In light of the reviews (below this email), we would like to invite the resubmission of a significantly-revised version that takes into account the reviewers' comments. 

We cannot make any decision about publication until we have seen the revised manuscript and your response to the reviewers' comments. Your revised manuscript is also likely to be sent to reviewers for further evaluation.

Sincerely,

Alain Kohl

Guest Editor

Philippe Desprès

Deputy Editor

Reviewer's Responses to Questions

**Key Review Criteria Required for Acceptance?**

**Methods**

-Are the objectives of the study clearly articulated with a clear testable hypothesis stated?

-Is the study design appropriate to address the stated objectives?

-Is the population clearly described and appropriate for the hypothesis being tested?

-Is the sample size sufficient to ensure adequate power to address the hypothesis being tested?

-Were correct statistical analysis used to support conclusions?

-Are there concerns about ethical or regulatory requirements being met?

Reviewer #1: -Are the objectives of the study clearly articulated with a clear testable hypothesis stated? The hypothesis and objectives of the study are not clear. The main concern is the lack of information in the methodology and some results that are missing. 

-Is the study design appropriate to address the stated objectives?

-Is the population clearly described and appropriate for the hypothesis being tested? the hypothesis is not clear

-Is the sample size sufficient to ensure adequate power to address the hypothesis being tested? yes

-Were correct statistical analysis used to support conclusions? 

-Are there concerns about ethical or regulatory requirements being met? no

Reviewer #2: The study design was clearly articulated to reach the objectives. Although the sample size it was small, the population it was well defined, clearly described and according to ethical and regulatory requirements needs. The multivariated statistical analysis is a powerfull method indicated to evaluate metabolomic data. Despite of the small size population, the statistical evaluation provided a good models allowing the authors suggest testable hypothesis revealing important biochemical mechanisms involved in TBEV infections. However these hypothesis was not tested in this work.

Reviewer #3: Are the objectives of the study clearly articulated with a clear testable hypothesis stated? - objectives are clearly stated, a clear hypothesis could be improved 

Is the study design appropriate to address the stated objectives? The study design is appropriate and address the objectives

Is the population clearly described and appropriate for the hypothesis being tested? Yes. It would be good to know for the patient data – how many were on drugs/medications during serum collection? This might impact the observations which is fine, but should be discussed. 

Is the sample size sufficient to ensure adequate power to address the hypothesis being tested? sample size is acceptable

Were correct statistical analysis used to support conclusions? Yes

Are there concerns about ethical or regulatory requirements being met? No concerns

Modifications needed:

Please include pathway analysis parameters in the M&M

**Results**

-Does the analysis presented match the analysis plan?

-Are the results clearly and completely presented?

-Are the figures (Tables, Images) of sufficient quality for clarity?

Reviewer #1: -Does the analysis presented match the analysis plan? no

-Are the results clearly and completely presented? no

-Are the figures (Tables, Images) of sufficient quality for clarity?

Reviewer #2: The rational pipeline to identify the detected features was good and the chemometric analysis was performed using the best softwares avaliable until this moment. The results were clearly and completely presented in tables and images adequated for this kind of experiments.

Reviewer #3: Does the analysis presented match the analysis plan? Yes

Are the results clearly and completely presented? No. The acquisition of the data are appropriately done. However, the results need far more description and inclusion of data.

Are the figures (Tables, Images) of sufficient quality for clarity? No. Most figure legends are not adequate and do not define the legends of the plots appropriately. Some of the figure legends are in chinese. 

Specifically, 

Table 1 should be a supplementary table and should include normalized abundance or log 2 fold change for each molecular feature, for targeted MS/MS verification, MSI levels should be indicated (see PMID: 24039616). A comprehensive list of the metabolites/lipids observed including data for 'unidentified' molecular features should be included. 

Table 2 – legend needs improvement for general readership (ie: what is VIP)

Figure 1 – please define legend of PCA plots (ie: what does T3-POS-H mean, what does DMODX circle size refer to?)

Figure 2 – for both A and B define colors. 

Figure 3 – text above the boxes are not legible or understandable to general readership (ie: what is TAG 597_group_monoisotopic). The text above the boxes should be 'putative or validated metabolite or lipid identifiers'

Figure S1. 

Fig (A): correct explanation

Fig (B): In positive mode, loss of 141 Da (716-575=141) is the characteristic neutral loss of PE

Fig (C): In negative mode, m/z 255.2327 corresponds to (C16:0) and m/z 279.2319 corresponds to (C18:2) are the daughter/product ions of m/z714.5095

MS/MS validated metabolites/lipids - representative spectra should be included - not just a PE example as in S1.

**Conclusions**

-Are the conclusions supported by the data presented?

-Are the limitations of analysis clearly described?

-Do the authors discuss how these data can be helpful to advance our understanding of the topic under study?

-Is public health relevance addressed?

Reviewer #1: -Are the conclusions supported by the data presented? no

-Are the limitations of analysis clearly described? no

-Do the authors discuss how these data can be helpful to advance our understanding of the topic under study? no

-Is public health relevance addressed?

Reviewer #2: Since TBEV is an increasing public health problem, this work represents an initial perspective of biochemical pathways involved in this kind of infection. Although this ilness was already evaluated by transcriptomic an proteomics, it is the first report revealing metabolic changes triggered by TBEV infection. The metabolites identified in this study provide insights into the fundamental metabolic mechanisms and pathways involved in TBEV infection and pathogenesis in humans being relevant from a public health perspective. The authors are aware of the limitations regarding to the small number of patients evaluated and described this point clearly in the discussion. The discussion was well developed by the authors in face of the data presented and contributed for the advance in the understandign of the TBE at biochemical level.

Reviewer #3: Conclusion could be supported by the data, however, more details of the data needs to be included in the manuscript. 

Are the limitations of analysis clearly described? Yes

Do the authors discuss how these data can be helpful to advance our understanding of the topic under study? Yes

Is public health relevance addressed? Yes

**Editorial and Data Presentation Modifications?**

Reviewer #1: (No Response)

Reviewer #2: Minor revision

line114 … antibodies for TBEV, and the ratio of alanine aminotransferase to aspirate (change aspirate to aspartate) transaminase ( ALT/AST). 

Line158... The identification of lipids molecular (change “lipids molecular” to “lipid molecules”) and metabolites followed an established strategy.

Line 280 ...and pathological changes, as well as impairing barrier function impaired (delet this word)

Reviewer #3: See results section

**Summary and General Comments**

Reviewer #1: (No Response)

Reviewer #2: Despite of TBEV virus is the most widespread arbovirus and a major public health threat in Europe and Asia, there are no metabolomic studies about that and little is known about the molecular mechanism underlying infection. YanDan Du et al., showed a system level comparison of TBEV patients serum metabolome and lipidome with a health control group. The authors have found physiological changes including Twenty-eight metabolites and 14 lipids significantly altered. Major perturbed metabolic pathways included amino acid metabolism, lipid and oxidative stress metabolism (lipoprotein biosynthesis, arachidonic acid biosynthesis, leukotriene biosynthesis and sphingolipid metabolism), phospholipid metabolism and triglyceride metabolism, is potentially involved in the acute inflammatory response and immune regulation. The authors suggest a potential utility of this distinct temporal serum metabolome changes as prognostic markers. In previous studies the authors already characterized serum cytokines, and measured the levels of lymphocyte subsets levels in tick-encephalitis patients and healthy controls to elucidate the relationship between those two factors and immune pathogenesis. I agree that metabolomics could complement previous “omic” studies revealing how the human organism responds, biochemically and physiologically to TBEV, as well as the associated cellular response and molecular mechanisms involved. So, I would like to suggest that the authors integrate this metabolomic data with omics data published previously by other authors in order to check if the data corroborates to the hypothesis argued by the authors. The major drawback of this work is the small number of samples as the authors mention in the discussion section. Although the hypothesis stated by the authors makes sense, they did not tested it. Some metabolites suggested to be a good biomarker could be tested by target metabolomics using a triple quadrupole comparing with a standards and after tested in the serum of another population or, at least, in another batch of analysis. I disagree when the authors arguee that some metabolites found in the serum would be related to the virus replication. In my opinion, it is difficult correlates metabolites founded in serum with process that occurs into the intracellular environment. I think that the metabolites found in the serum is derived to changes that the virus triggers in the physiology of the host in a general mode. To confirm this hypothesis stated by the authors, they could perform a metabolomic analysis using an in vitro model of infection.

Reviewer #3: The manuscript describes a metabolomics/lipidomics analysis of TBEV patient sera compared to healthy controls and reveals important and scientifically interesting information and therefore, should be published. However, improved data inclusion, improved figure legends and figures and data tables are required prior to acceptance. 

Additional general comments:

Entire manuscript - English edits necessary

Introduction: Vaccine is available for TBEV

Line 114, pg 8 - ALT/AST – aspartate transaminase

PLOS authors have the option to publish the peer review history of their article (what does this mean?). If published, this will include your full peer review and any attached files.

Reviewer #1: No

Reviewer #2: Yes: Juliano Simoes de Toledo

Reviewer #3: No
---

## [Decision Letter · Decision Letter 1]

24 Nov 2020

Dear Dr YiQing Niu

Thank you very much for submitting your manuscript "Insights into the molecular basis of tick-borne encephalitis from Multiplatform Metabolomics" for consideration at PLOS Neglected Tropical Diseases. As with all papers reviewed by the journal, your manuscript was reviewed by members of the editorial board and by several independent reviewers. The reviewers appreciated the attention to an important topic. Based on the reviews, we are likely to accept this manuscript for publication, providing that you modify the manuscript according to the review recommendations. 

Please respond to the comments by reviewers 1 and 3 in particular, carefully. Clarity of figures and legends (figures 3/4) in particular should be improved.

Sincerely,

Alain Kohl

Associate Editor

Philippe Desprès

Deputy Editor

Please respond to the comments by reviewers 1 and 3 in particular, carefully. Clarity of figures and legends (figures 3/4) in particular should be improved.

Reviewer's Responses to Questions

**Key Review Criteria Required for Acceptance?**

**Methods**

-Are the objectives of the study clearly articulated with a clear testable hypothesis stated?

-Is the study design appropriate to address the stated objectives?

-Is the population clearly described and appropriate for the hypothesis being tested?

-Is the sample size sufficient to ensure adequate power to address the hypothesis being tested?

-Were correct statistical analysis used to support conclusions?

-Are there concerns about ethical or regulatory requirements being met?

Reviewer #1: The hypothesis may still be improved. In this study the authors used several approaches to separate the metabolites according the polarity of the compounds. This indicates that part of their objectives is to have high coverage in the metabolomics study. This should be mentioned in the introduction to make the objective of the study more clear. 

 The sample size is sufficient. However, the amount of samples was not provided in the methodology but in the results section. This information should be in the methodology. 

-Were correct statistical analysis used to support conclusions? There are some issues about the statistical analysis that still need to be clarified. See below. 

1) Did the authors performed correction for multiple hypothesis after the T-test? This is important to avoid false-positives. 

2) The authors are not clear about the amount of features in the QC that exhibit a relative standard deviation (RSD) lower than 10%. In the text, it seems that all features from the QCs injections exhibited a RSD<10%. Although, the majority of features could display a RSD<10% it is unlikely that all features presented such behavior. The authors should mentioned the percentage of features that presented a RSD<10%.

3) The authors mentioned that features with a p<0.05 displayed a significant difference between the samples. However, was the fold change (or log2FC) used as criteria to classify a feature as significant? Please clarify the criteria used to claim that a feature was statistical significant. 

-Are there concerns about ethical or regulatory requirements being met? no

Reviewer #2: The study design and the methods are appropriated to reach the objectives. The multivariated statistical analysis is a powerfull method indicated to evaluate metabolomic data and support the conclusions. Although the sample size it was small, the population it was well defined, clearly described and according to ethical and regulatory requirements needs according to the ethics committee of Inner Mongolia Forestry General Hospital. Despite of the small size population, the statistical evaluation provided a good models allowing the authors suggest testable hypothesis revealing important biochemical mechanisms involved in TBEV infections.

Reviewer #3: This study uses systems biology based metabolomics and lipidomics methods to identify prognostic biosignatures of TBEV. The methodology is very detailed, and the authors have taken great care to carry out the appropriate controls for analyzing and interpreting the data which provides great confidence in their readouts.

**Results**

-Does the analysis presented match the analysis plan?

-Are the results clearly and completely presented?

-Are the figures (Tables, Images) of sufficient quality for clarity?

Reviewer #1: -Does the analysis presented match the analysis plan? yes

-Are the results clearly and completely presented?The results are clear

-Are the figures (Tables, Images) of sufficient quality for clarity?the images present a sufficient quality

Reviewer #2: The results presented match to the study design proposed. The rational pipeline to identify the detected features was good and the chemometric analysis was performed using the best softwares avaliable until this moment. The results were clearly and completely presented in tables and images adequated for this kind of experiments.

Reviewer #3: The results are very informative, but requires some improvement in presentation. 

Fig 3 right panel - is unreadable and not useful. Can you present that data using an alternative graphic - may be using more dimensions?

Fig 4 - I did not find the figure legend. This figure legend needs to be very detailed to walk the reader through the hypotheses behind the pathway diagram. So while the figure is probably the most important take-away, some detailed descriptions are neccessary.

Lines 184-185 - Please correct accuracy if sentence: “PCA scores representing TBEV samples and healthy samples were obtained using the T3 column and BEH Amide column” – the PCA scores were not obtained using the columns. It was obtained using the data generated from the methods using T3 and BEH columns.

Also, none of the supplemental files were downloadable - these are critical to evaluate the reporting of the data. Please provide.

**Conclusions**

-Are the conclusions supported by the data presented?

-Are the limitations of analysis clearly described?

-Do the authors discuss how these data can be helpful to advance our understanding of the topic under study?

-Is public health relevance addressed?

Reviewer #1: (No Response)

Reviewer #2: TBE is an increasing public health problem and the data presented have public health relevance because represents an initial perspective of biochemical pathways involved in this kind of infection. Although this ilness was already evaluated by transcriptomic an proteomics, it is the first report revealing a comprehensive metabolic changes triggered by TBEV infection, including lipid analysis. The metabolites identified in this study provide insights into the fundamental metabolic mechanisms and pathways involved in TBEV infection and pathogenesis in humans being relevant from a public health perspective. The authors are aware of the limitations regarding to the small number of patients evaluated and described this point clearly in the discussion. The discussion was well developed by the authors in face of the data presented and contributed for the advance in the understanding of the TBE at biochemical level.

Reviewer #3: Conclusions drawn are valid and informative. Again, better presentation will communicate the conclusions better.

**Editorial and Data Presentation Modifications?**

Reviewer #1: minor modifications 

1) Lines 212-213. Is not correct to use the term expression when refer to a metabolite level. A metabolite is not expressed. 

2) In line 175, it was mentioned that “…The enrichment analysis was performed using a topology analysis that takes into account the positions of significant metabolites in metabolic pathways…” Actually, in the Metaboanalyst the pathway analysis combines enrichment and topology analysis to indicate which metabolic pathways are disturbed in a disease or experimental condition. Please rephrase this sentence to improve the explanation about pathway analysis in Metaboanalyst.

3) Line178. Which criteria in the pathway analysis were used to consider that a metabolic pathway was disturbed? Impact values closer to 1.0 indicate a more perturbed pathway and some works consider that when an impact value for a pathway is higher than 0.1 this indicate that the pathway is disturbed. Did the authors set a threshold for the pathway impact value?

Reviewer #2: (No Response)

Reviewer #3: (No Response)

**Summary and General Comments**

Reviewer #1: The majority of the concerns were addressed. However, some problems still remained (see above).

Reviewer #2: Despite of TBEV virus is the most widespread arbovirus and a major public health threat in Europe and Asia, there are no comprehensive metabolomic studies about that and little is known about the molecular mechanism underlying infection. YanDan Du et al., showed a system level comparison of TBEV patients serum metabolomics and lipidomics with a health control group. The authors have found physiological changes including Twenty-eight metabolites and 14 lipids significantly altered. Major perturbed metabolic pathways included amino acid metabolism, lipid and oxidative stress metabolism (lipoprotein biosynthesis, arachidonic acid biosynthesis, leukotriene biosynthesis and sphingolipid metabolism), phospholipid metabolism and triglyceride metabolism, is potentially involved in the acute inflammatory response and immune regulation. The authors suggest a potential utility of this distinct temporal serum metabolome changes as prognostic markers. In previous studies the authors already characterized serum cytokines, and measured the levels of lymphocyte subsets levels in tick-encephalitis patients and healthy controls to elucidate the relationship between those two factors and immune pathogenesis. The major drawback of this work is the small number of samples as the authors mention in the discussion section. Although the hypothesis stated by the authors makes sense, they did not tested it. Some metabolites suggested to be a good biomarker could be tested by target metabolomics using a triple quadrupole comparing with a standards and after tested in the serum of another population or, at least, in another batch of analysis.

Reviewer #3: (No Response)

PLOS authors have the option to publish the peer review history of their article (what does this mean?). If published, this will include your full peer review and any attached files.

Reviewer #1: No

Reviewer #2: Yes: Juliano Simões de Toledo

Reviewer #3: No
---

## [Editor Report · Decision Letter 2]

8 Jan 2021

Dear Dr YiQing Niu,

Thank you very much for submitting your manuscript "Insights into the molecular basis of tick-borne encephalitis from Multiplatform Metabolomics" for consideration at PLOS Neglected Tropical Diseases. As with all papers reviewed by the journal, your manuscript was reviewed by members of the editorial board and by several independent reviewers. The reviewers appreciated the attention to an important topic. Based on the reviews, we are likely to accept this manuscript for publication, providing that you modify the manuscript according to the review recommendations. 

The authors make reference in line 95 to a Masters thesis in Chinese and supplementary figures. We do not see why this is included at the final paragraph of the Introduction, and the relevance to the paper. All data should be referring to figures in the paper, not a Masters thesis published elsewhere, and their relevance and contribution clearly stated. We suggest removing this, and be clear what part of the Results section data in supplemental figures relate to.

Sincerely,

Alain Kohl

Associate Editor

Philippe Desprès

Deputy Editor

The authors make reference in line 95 to a Masters thesis in Chinese and supplementary figures. We do not see why this is included at the final paragraph of the Introduction, and the relevance to the paper. All data should be referring to figures in the paper, not a Masters thesis published elsewhere, and their relevance and contribution clearly stated. We suggest removing this, and be clear what part of the Results section data in supplemental figures relate to.
---

## [Editor Report · Decision Letter 3]

23 Jan 2021

Dear Dr YiQing Niu,

We are pleased to inform you that your manuscript 'Insights into the molecular basis of tick-borne encephalitis from Multiplatform Metabolomics' has been provisionally accepted for publication in PLOS Neglected Tropical Diseases.

A number of mistakes in the Author Summary should be addressed with the publishers (see comment below).

Best regards,

Alain Kohl

Associate Editor

Philippe Desprès

Deputy Editor

The paper can now be accepted, but authors should work with the publisher to correct mistakes in the revised Author Summary.

---

## [Editor Report · Acceptance letter]

24 Feb 2021

Dear Niu,

We are delighted to inform you that your manuscript, "Insights into the molecular basis of tick-borne encephalitis from Multiplatform Metabolomics," has been formally accepted for publication in PLOS Neglected Tropical Diseases.

Best regards,

Shaden Kamhawi

co-Editor-in-Chief

Paul Brindley

co-Editor-in-Chief
